# Mechanisms of Prostate Cancer Cells Survival and Their Therapeutic Targeting

**DOI:** 10.3390/ijms24032939

**Published:** 2023-02-02

**Authors:** Tomislav Pejčić, Zoran Todorović, Siniša Đurašević, Lazar Popović

**Affiliations:** 1Faculty of Medicine, University of Belgrade, 11000 Belgrade, Serbia; 2Clinic of Urology, University Clinical Centre of Serbia, 11000 Belgrade, Serbia; 3University Medical Centre “Bežanijska kosa”, University of Belgrade, 11000 Belgrade, Serbia; 4Faculty of Biology, University of Belgrade, 11000 Belgrade, Serbia; 5Faculty of Medicine, University of Novi Sad, 21000 Novi Sad, Serbia; 6Medical Oncology Department, Oncology Institute of Vojvodina, 21000 Novi Sad, Serbia

**Keywords:** prostate, cancer, therapy, antiandrogen resistance, new drugs

## Abstract

Prostate cancer (PCa) is today the second most common cancer in the world, with almost 400,000 deaths annually. Multiple factors are involved in the etiology of PCa, such as older age, genetic mutations, ethnicity, diet, or inflammation. Modern treatment of PCa involves radical surgical treatment or radiation therapy in the stages when the tumor is limited to the prostate. When metastases develop, the standard procedure is androgen deprivation therapy, which aims to reduce the level of circulating testosterone, which is achieved by surgical or medical castration. However, when the level of testosterone decreases to the castration level, the tumor cells adapt to the new conditions through different mechanisms, which enable their unhindered growth and survival, despite the therapy. New knowledge about the biology of the so-called of castration-resistant PCa and the way it adapts to therapy will enable the development of new drugs, whose goal is to prolong the survival of patients with this stage of the disease, which will be discussed in this review.

## 1. Introduction

PCa is the second most common cancer in the world, with nearly 400,000 deaths annually. Multiple factors are involved in the etiology of PCa, such as older age, genetic mutations, ethnicity, diet, or inflammation. Today’s “epidemic” increase in the incidence of PCa may be linked to the change in the diet of *Homo sapiens*, which happened relatively recently with the development of animal husbandry—animal fat, especially saturated, can contribute to PCa progression, while unsaturated fatty acids (FA) abundant in fish and vegetable oils reduce PCa risk [1]. PCa is rare in Eastern countries where diet is dominated by plants and very common in the West where the Western diet predominates [2].

Modern treatment of PCa is most effective when it is detected at the local stage, i.e., when the tumor is limited to the prostate, when radical surgical treatment or radiation therapy is carried out. At the time of diagnosis of PCa, 73% of patients have disease limited to the prostate, while 14% have metastases in regional lymph nodes, and 7% have distant metastases (Cancer Stat Facts: Prostate Cancer. National Cancer Institute. Available at https://seer.cancer.gov/statfacts/html/prost.html, accessed on 2 November 2022). In PCa metastases, genetic damage continues, which leads to greater aggressiveness of the tumor and an increase in its metastatic potential. Special adaptation of tumor cells in metastases occurs during androgen deprivation therapy (ADT); malignant cells then go through a series of adaptations to survive in very low levels, the so-called castration level, of serum testosterone. Therefore, this stage of disease is called castration-resistant PCa (CRPCa). Various new processes are then observed in PCa cells, such as amplifications and mutations on the androgen receptor (AR), amplifications on the regulatory enzyme in steroid synthesis, CYP17A1, which can activate de novo synthesis of androgens. In addition, the so-called rescue pathways are activated; these pathways include growth factors and activate AR signaling that is completely independent of androgens and other steroids.

The standard primary therapy for metastatic prostate carcinoma (mPCa) is still ADT, that is, surgical or medical castration. However, after ADT, after a period of remission, CRPCa develops and continues to grow and progress. Therefore, the development of new drugs is important, such as new AR blockers, CYP17A1 blockers, inhibitors of the reparative enzyme PARP, and others.

## 2. Physiology and Pathophysiology of the Prostate

The prostate is the most widespread accessory sex gland in mammals. The main function of the prostate is the production of secretions that enable the function and survival of spermatozoa. PCa, which is one of the most common cancers in humans today, is exceedingly rare in most mammals, including herbivorous hominids.

The structure of the prostate consists of stroma, epithelium, and secretory ducts. The stroma consists of cells (fibroblasts, endothelial cells of capillaries and lymphatic vessels, and smooth muscle cells) and extracellular matrix. The epithelium of the prostate consists of cells (secretory, neuroendocrine, intermediate, basal, and stem cells). During embryonic development, the development of the prostate in puberty, and throughout adulthood, there is a constant interaction between the stroma and the epithelium, during which androgenic and estrogenic hormones stimulate the activity of the stroma which further stimulates the growth and proliferation of the epithelium and its secretory activity (Figure 1) [3,4,5].

The prostate secretes numerous non-peptide (citrate, zinc, polyamines, cholesterol) and peptide substances (prostate-specific antigen—PSA, glandular kallikrein—hK2, prostatic acid phosphatase—PAP) [6,7,8,9]. The hormonal function of the prostate is paracrine, acting on neighboring cells (fibroblast growth factor—FGF, epidermal growth factor—EGF) [10], and intracrine, acting inside the cell (5α-dihydrotestosterone—5α-DHT) [11]. Direct cell communication is achieved through cell adhesion molecules (CAM) [12]. The PSA enzyme has a physiological function only in the vagina, where it liquefies the seminal coagulum and thus enables active sperm motility. Prostate secretion rich in PSA molecules constantly leaks from the prostate duct into the urethra, where it is flushed out with urine. The determination of PSA and its derivatives in the serum today represents the beginning of the diagnosis of prostate disease [13,14,15,16,17,18,19,20].

## 3. PCa Etiology

According to 2020 GLOBOCAN data, PCa is the second most common cancer among men and the fifth leading cause of cancer death, with approximately 1.4 million new cases and approximately 375,000 deaths worldwide (Figure 2). The incidence is three times higher in developed than in underdeveloped countries and varies from 6.3 to 83.4 per 100,000 men in different regions, with the highest rates in Northern and Western Europe, the Caribbean, Australia/New Zealand, North America, and South Africa, and the lowest in Asia and North Africa [21].

PCa is diagnosed in only 2% of cases in people under the age of 50, and over 60% of patients are diagnosed after the age of 65 [22]. In 2018, the Task Force on Chronic Disease in the USA proposed screening for PCa in men aged 55 to 69 years [23]. In addition, patients with well, moderately, and poorly differentiated tumors had 10-year CSS rates of 91%, 90%, and 74%, respectively [24]. The overall PCa mortality based on studies from 185 countries is shown in the Figure 2 [21]. Roughly, active treatment is indicated in patients with local disease and life expectancy over 10 years. Patients who have not received local treatment have a cancer specific survival (CSS) rate of 82–87% after 10 years [21].

Risk factors for PCa include aging, a family history of PCa, and genetic mutations, such as *BRCA1* and *BRCA2*. African Americans in the USA have the highest incidence rate globally, which is explained by African ancestry and multiple environmental factors, such as hypovitaminosis D. About 15% of PCas are thought to be hereditary. Thus, the risk of developing PCa is 2× higher if the father had PCa, 3× if the brother had PCa, and if both father and brother had PCa, as much as 5 times [25,26].

### 3.1. Lifestyle Influence

Changes in lifestyle have a substantial effect on prostate carcinogenesis [27]. Low physical activity typical of sedentary life in the West, along with a high-fat diet, leads to obesity and chronic systemic inflammation, which may be involved in PCa development via immune-related processes [28]. Moreover, as the modern man’s longevity has expanded dramatically, so has the duration of the influence of oxidative stress on the formation of genetic damage and mutations. It is believed that long-term hormonal stimulation, persistent oxidative stress and the resulting inflammation, and cumulative genetic damage are responsible for the significant increase in the prevalence of PCa and breast cancer [29]. Chronic inflammation accompanied by tissue repair and cell hyperproliferation, either from infection, a high-fat diet, or other causes, also contributes to the development of PCa [30]. For example, proliferative inflammatory atrophy is a common finding in prostate tissue specimens [31]. In addition, studies have proven a link between obesity and an increased risk of PCa [32,33,34]. In obese people, the concentration of leptin, estradiol, insulin, and free insulin-like growth factors 1 (IGF-1) increased in the serum, and the concentration of free testosterone and adiponectin decreased [35,36,37,38]. Leptin is a peptide hormone produced in adipocytes, which affects appetite suppression and energy consumption; there is evidence that leptin plays a role in the development of advanced PCa [39,40]. Insulin-like growth factors (IGFs) are peptide hormones that affect basic processes in prostate cells [41], such as stimulated proliferation, migration, and inhibited apoptosis [42].

Nutritional factors appear to play a role in the promotion of PCa. There are a lot of data offered by paleontology, evolution, and hominid structure, which prove that Homo sapiens was an obligate herbivore for hundreds of thousands of years, and that only relatively recently, with the development of animal husbandry, did it drastically increase its intake of meat and nutrition [43]. According to WHO data from 2012, the incidence of PCa in Central Asia was 4.5, in Japan, 31.2, and in Australia, over 100 [44]. In addition to the genetic component, it is believed that the diet, which, in Asia, is based on plant foods, with little animal protein and fat, is also responsible for the lower risk [45,46]. Epidemiological studies show that Indians who migrate to the USA have a 10-fold higher incidence of PCa than Indians in India (47.4 vs. 4.6) [47]. Among young Asians in the USA, there is a jump in the incidence of PCa, but not in the older population, which maintains a traditional eating style [48]. Additionally, Japanese men residing in Japan who prefer the “Western-style diet pattern” (red meat, potatoes, and high-fat dairy products) have a higher chance of developing PCa than men who prefer the “prudent diet pattern” (vegetables, fruits, and fish) [49].

Zinc is an important nutritional factor. Its deficiency is linked with PCa occurrence and progression [50]. The zinc ion is a cofactor of several cell enzymes, including antioxidative (cooper zinc superoxide dismutase, CuZnSOD). Its tissue accumulation depends both on the nutritional availability and the activity of two transporter families, one that regulates cellular zinc inflow from the extracellular milieu (the ZIP or Zrt/IRT-like proteins) and another controlling zinc cellular outflow and redistribution into the mitochondria and lysosomes (ZnT or Zn transporter). The loss of zinc accumulation, either due to a nutritional deficit or a decrease in transporter activity, as is the case within PCa [51], has manifold consequences, including decrease in citrate production. Citrate is one of the main physiological components of prostatic fluid, with the production based on a specific metabolic pathway not existent in other cells. During glycolysis, glucose is transformed into pyruvate, which is further oxidized into acetyl-CoA. In non-prostatic tissue, acetyl-CoA may react with oxaloacetate, thus producing citrate which enters the Krebs cycle. However, in prostate cells, the activity of the aconitase 2, the mitochondrial enzyme necessary for the first step of citrate oxidation, is inhibited by zinc ions. This blocks the entry of citrate into the Krebs cycle, leading to its prostate accumulation and further secretion in the semen [52]. Citrate has high affinity for zinc, magnesium, and calcium ions. Since calcium affects sperm count, motility, morphology and volume, variations in zinc accumulation, and thus citrate secretion, alter male fertility [53]. Zinc is also involved in other biological processes such as cell division, intracellular signaling, apoptosis, cell invasion, and migration. Induced zinc accumulation in PCa cells leads to a marked inhibition of cell growth mediated by stimulation of p21 [54], and reduced expression of tumorigenic cytokines VEGF, IL-6, IL-8, and MMP-9 [50]. Conversely, depletion of intracellular zinc increases VEGF, IL-6, and IL-8 expression in PCa cells via the NF-kB-dependent pathway [55], and increases expression and phosphorylation of PKB/Akt and Mdm2, and decreases p53 and p21 levels in normal and cancer prostate cells [56].

Another nutrient the deficiency of which has been linked with the PCa is vitamin D (1,25-dihydroxy-Vitamin D3), a member of the steroid hormone superfamily. In humans, vitamin D from food and during exposure to sunlight is converted into active vitamin D in the skin. Vitamin D binds to the vitamin D receptor (VDR) in prostate cells, which is present in both normal and malignant prostate epithelium [57]. Stimulation of VDR leads to cell cycle arrest and inhibition of malignant cell migration [58,59,60,61]. Studies show that men with vitamin D deficiency have a higher incidence of PCa, and these are men who live in areas with few sunny days, African Americans, and older people [62,63]. However, it should be underlined that recent studies question the role of vitamin D in PCa prevention [62,64]. 

### 3.2. Hormonal Influence

Regulation of prostate function and growth is mainly conducted by androgenic and estrogenic hormones and growth factors. The major circulating androgen, testosterone, is synthesized in the testes from acetate and cholesterol under the stimulation of luteinizing hormone (LH). In prostate cells, more than 90% of testosterone is irreversibly converted to 5α-DHT via the enzyme 5-alpha-reductase (5αR). The 5α-DHT molecules and the remaining testosterone molecules then bind to the androgen receptor. This is followed by dimerization or formation of pairs of AR–androgen complexes which enter the nucleus and bind to androgen responsive elements on DNA. It is known that the majority of PCas initially respond well to androgen deprivation and that the AR plays a decisive role in the development of castration-resistant disease. Two large studies, PCPT (Prostate Cancer Prevention Trial) and REDUCE (Reduction by Dutasteride of Prostate Cancer Events), have shown that drugs that inhibit 5αR reduce the incidence of PCa by 25–30% [65].

Estrogen hormones in healthy, young men are mainly produced in peripheral tissues, mostly in adipose tissue, by the conversion of androstenedione (ASD) to estrone (E1) and testosterone to estradiol (E2), and only a small amount is produced in the testicles. This conversion is carried out by the enzyme cytochrome P450 (CYP) 19, or aromatase [66]. Estrogens are especially accumulated in the stroma of the prostate, which happens especially in old age [67], acting on the human prostate via interaction with two receptor subtypes, ERα and Erβ [68]. ERα is found primarily in the stroma, and ERβ in both the stroma and the epithelium [69]. Stimulation of ERα stimulates cell proliferation, while stimulation of ERβ counteracts these effects [70]. Estrogens have always been considered as protective factors for the development of PCa and have long been used in the treatment of advanced PCa. This effect of estrogen is due to negative feedback on the hypothalamic–pituitary–gonadal axis. However, today it is known that estrogens in the prostate can act as procarcinogens [71], with the action mainly carried out via ERα [72]. As PCa progresses, ERα receptors in basal and stromal cells become more numerous, while at the same time, the number of ERβ receptors decreases, which inhibits stromal–epithelial interaction and proliferation and activate apoptosis [67,73,74,75]. Animal experiments show that it is impossible to induce PCa in mice lacking ERα (alphaERKO) and aromatase enzyme (ArKO) [76,77]. Hence, locally produced estrogens in the prostate and malignant tissue, and not the level of estrogen in the serum, are of key importance for the growth and progression of PCa [78].

Growth factors (GF) are small peptide molecules that stimulate and sometimes inhibit the process of cell division and differentiation. Cells that respond to GF stimulation have receptors specific for a particular GF on their surface [79]. The balance between cell proliferation and death is dictated by complex interactions between GF and steroid hormones. FGF (fibroblastic GF), VEGF (vascular endothelial GF) and IGF are thought to promote cell proliferation in BPH, and 5α-DHT enhances their effects [80]. The transforming factor GF-β (TGF-β1) performs several essential functions: it stimulates mitosis in fibroblasts, inhibits the proliferation of epithelial cells, and induces apoptosis. Increased expression of TGF-β1 on stromal cells is associated with an increase in the stromal compartment [81]. Interestingly, TGF-β can act as an inhibitor of normal growth and early-stage disease and a promoter of advanced PCa [82].

### 3.3. Metabolic Influence

#### 3.3.1. Glucose Metabolism Remodulation

There are several metabolic processes whose changes may contribute to PCa development and progression. PCa cells have a hybrid glycolytic/oxidative phosphorylation (OXPHOS) phenotype. One route for the transition from OXPHOS to aerobic glycolysis involves lactate dehydrogenase (LDH) isoenzymes [83]. LDH isoform A (LDH-A) preferentially converts pyruvate to lactate, while LDH isoform B (LDH-B) can eventually oxidize lactate back to pyruvate. Hence, decreased LDH-A phosphorylation and increased LDH-B expression may switch OXPHOS to aerobic glycolysis. Another route is the activity of mitochondrial pyruvate carrier (MPC), which transfers pyruvate into mitochondria [83]. Since mitochondrial enzymes that metabolize pyruvate are physically separated from its cytosolic pool, the mitochondrial metabolism of pyruvate, and thus switch from OXPHOS to aerobic glycolysis, relies on MPC. Bader et al. (2019) demonstrated that AR-regulated MPC transcription is elevated in primary PCa and is linked with a poor prognosis [84].

Another important glucose pathway is the pentose phosphate pathway (PPP), by which cells, using glucose-6-phosphate, generate NADPH and the precursors for nucleotide synthesis. As a result of AR signaling, the expression of the PPP rate-limiting enzyme, glucose-6-phosphate dehydrogenase (G6PDH), is increased in PCa cells, boosting G6PDH, NADPH, and ribose synthesis [85].

It is known that the OXPHOS phenotype may trigger the adenine monophosphate-activated protein kinase (AMPK)/proliferator-activated receptor gamma coactivator 1-alpha (PGC-1α) cascade [86,87]. AMPK is one of the central regulators of cellular and organismal metabolism activated when intracellular ATP production decreases [88], whereas PGC-1α is a transcriptional coactivator that plays a crucial role in mitochondrial biogenesis and function [89]. Ros et al. (2012) described PRKAB1 and PFKFB4 genes as crucial for PCa cell survival [90]. PRKAB1 encodes an AMPK regulatory subunit that inhibits ATP consumption. PFKFB4 encodes an isoform of the glycolytic enzyme phosphofructokinase 2. Due to the enzyme’s central role in regulating glycolysis and antioxidant generation, higher PFKFB4 mRNA levels were observed in metastatic PCa compared with localized tumors. In addition, expression of human serine/threonine kinase 11 (also known as liver kinase B1, LKB1), which directly activates AMPK, was markedly diminished in high-grade prostatic intraepithelial neoplasia lesions and entirely abolished in adenocarcinomas [91].

In controlling oncogenic processes, the interaction of the LKB1-AMPK axis with the phosphoinositide 3-kinase III (PI3K) and mammalian target of rapamycin (mTOR) pathways is critical [87]. Being necessary for the promotion of growth and proliferation of adult stem cells, the PI3K/Akt/mTOR pathways are directly related to cancer [92]. Several known factors enhance PI3K/Akt activity, including EGF, Sonic Hedgehog, IGF-1, insulin, and CAM [93,94,95,96]. On the other hand, phosphatase tensin homolog (PTEN) and other factors have an antagonistic effect on this pathway [97]. PTEN is a natural inhibitor of the PI3K/Akt pathway and its function is to limit cell proliferation and help prevent cancer development [98]. Therefore, PTEN mutations represent a critical element in carcinogenesis [99]. Highly relevant to therapeutic development, AR blockade subsequently activates PI3K/Akt/mTOR pathways [100], offering a rationale for their therapeutic targeting [101,102,103]. In addition, AR mutations, amplification, and overexpression also represent an important element in prostate carcinogenesis [104,105].

#### 3.3.2. Lipid Metabolism Remodulation

Lipids are used as an energy source by the mitochondrial free fatty acids (FFA) β-oxidation, with the FAT/CD36 translocase as the primary FA transporter. The CD36 gene is typically amplified in PCa and linked to a poor patient prognosis. Further, CD36 acts as a receptor, initiating the signal transduction pathways of SRC family kinases, mitogen-activated protein kinases, and reactive oxygen species. Oxidized low-density lipoproteins, beta-amyloid peptide, microbial diacylglycerides lipoteichoic acid from Staphylococcus aureus, and mycoplasma macrophage-activating lipopeptide-2 are some of the CD36 ligands, which may explain its oncogenic potential. The tumor FA intake, lipid production, and oncogenic lipid signaling pathways are all inhibited when CD36 is blocked, limiting cancer progression [106,107].

The FA distribution within cells is regulated by the FA-binding proteins (FABPs). FABPs may have a role in the metastasis of other malignancies, including PCa. For example, squamous cell carcinoma metastases frequently involve amplification of the 8q21 region of the genome, which contains a cluster of FABP members. Peroxisome proliferator-activated receptors (PPARs) are FA-activated nuclear receptors that regulate lipid and glucose metabolism to maintain energy homeostasis [108]. By transferring FA to PPAR β/δ and γ to [50], FABPs activate PPARs’ downstream effectors, which may promote cancer epithelial–mesenchymal transition, angiogenesis, migration, and invasion [109,110].

The AR regulates the transcription of several lipogenic and cholesterogenic enzymes overexpressed in PCa, such as ATP-citrate lyase, acetyl-CoA carboxylase, FA synthase (FASN), stearoyl-CoA desaturase-1, 3-hydroxy-3-methylglutaryl-CoA reductase, and squalene epoxidase [111], with the transcription factors called sterol regulatory element-binding proteins (SREBPs) as a mediator in this process [112]. To be activated, an SREBP demands cleavage from the endoplasmic reticulum (ER) membrane in a two-step process in the Golgi apparatus by the SREBP cleavage-activating protein (SCAP). Because the SCAP–SREBP complex is kept at the rough ER when cholesterol is present, the activity of SREBP is dependent on the sterol intracellular content. SREBPs are activated when PI3K/AKT/mTOR is upregulated and PTEN is lost (both of which are common genetic abnormalities). Moreover, SREBP1 has been linked to PCa cell proliferation, migration, and invasion via activation of lipogenesis, increased generation of reactive oxygen species, and NADPH oxidase 5 expression. SCAP expression is induced by AR, while AR expression is enhanced by SREBP1. Because of this feedback loop formed by AR and SREBP1, gene expression can be maintained indefinitely [113]. Inhibition of FASN activity in castration-resistant cancer cell lines results in a decrease in tumor development, increased apoptosis, altered lipidome, decreased lipid storage, and increased AR expression [114]. Lipid droplets within high-grade and metastatic human PCa contain an abnormal build-up of esterified cholesterol, which is not present in normal prostate cells. Cholesterol accumulation is associated with the downregulation of sterol efflux and the upregulation of lipid synthesis and uptake from the circulation [115]. The use of statins (inhibitors of HMG-CoA reductase, a rate-limiting enzyme in cholesterol synthesis) has been shown to limit the proliferation, invasion, and migration of PCa cells and to trigger their programmed death [116]. HMG-CoA reductase overexpression is correlated with a poor prognosis; moreover, this enzyme is overexpressed in enzalutamide-resistant PCa cells and knocking it out can re-establish sensitivity to the drug. In both in vivo and in vitro models, PCa cell growth was suppressed when simvastatin and enzalutamide were used together. However, simvastatin reduced AR expression in enzalutamide-resistant cancer cells alone or in combination with enzalutamide [117,118,119].

#### 3.3.3. Amino Acid Metabolism Remodulation

Krebs cycle and lipogenesis intermediates are mostly synthesized from glutamine. Glutamine is transported into cells from the extracellular milieu by micropinocytosis or the solute carrier family of transporters [120,121]. Inside the cell, glutaminase GLS1 converts glutamine to glutamate, which is further converted into α-ketoglutarate (α-KG) by glutamate dehydrogenase or glutamic oxaloacetic transaminase. DNA demethylating enzymes and histone demethylases with Jumonji domains require α-KG as a cofactor [122,123]. Both the glutamine transporter ASCT2 (SLC1A5) and the glutaminase GLS1 are overexpressed in PCa cells. Many pathways, including AR, MYC, and mTOR, have been linked to these occurrences [124]. Glutamine also helps the tumor develop by donating nitrogen to other metabolites, including amino acids and nucleotides. Glutamine metabolism is a promising therapeutic target because of its pivotal role in developing PCa [125]. When glutamate sources are few, cancer cells can obtain glutamate from N-acetyl-aspartyl-glutamate (NAAG) [126]. In recent investigations, plasma NAAG concentrations correlate with tumor size in various cancers, while glutamate levels appear to correlate with the Gleason score [127].

Cancerous prostate tissue has a higher concentration of tryptophan (Trp) than healthy prostate tissue. The conversion of Trp into kynurenine (Kyn) is mediated by indoleamine-2,3-dioxygenase 1 (IDO1) and tryptophan 2,3-dioxygenase, with the IDO1 expression being regulated by IFN-γ and TNF-α. Kyn appears to play a role in immunological and cancer cells through its interaction with the aryl hydrocarbon receptor (AhR) in the cytoplasm. AhR enters the nucleus after interacting with Kyn, promoting the expression of genes involved in tumor cell migration and immune evasion. Renal and bladder malignancies have been linked to the Trp/Kyn pathway [128,129]. 

Sarcosine is an N-methyl glycine derivative whose expression increases with the transition from benign tissue, via localization, to metastatic malignancy. Transfer of a methyl group from S-adenosyl methionine to glycine is catalyzed by glycine-N-methyltransferase (GNMT). Greater GNMT expression was recently described in PCa cells compared to normal tissues, and a correlation was shown between higher enzyme levels and shorter times spent disease-free [130]. According to a prior study, patients with PSA range < 4 ng/mL have a stronger predictive value for serum sarcosine than for PSA. In addition, sarcosine levels were positively related to low, intermediate, and high-grade malignancies [131,132].

#### 3.3.4. The Role of Autophagy

One of the specific roles of PI3K/mTOR pathways in PCa is their involvement in the regulation of autophagy, a homeostatic process involved in several cell activities, including tissue growth, differentiation, metabolic modulation, and cancer development [133]. Activation of autophagy is essential for cancer cell survival once tumor mass has been established, despite autophagy’s role as a tumor suppressor in non-tumor cells and during the early phases of tumor cell growth. Cancer cells have a greater metabolic rate, which is supposed to mean that they use more energy and more building blocks. In order to survive metabolic deprivation, they increase autophagic flux and become reliant on external energy sources [134].

The AuTophaGy related 1 /Unc-51-like kinase-1 (ATG1/ULK1) complex and the PI3K-III work together to trigger autophagy in response to the cell’s energy needs. The following step is the nucleation of phagophores which phagocytize intracellular cargos, generating double-membranous structures called autophagosomes. The content of autophagosomes is destroyed to liberate amino acids and other metabolic components when they fuse with lysosomes to produce autolysosomes [135]. Multiple genes and enzymes work together to control autophagy and autophagosome production. Beclin-1, a protein expressed by the BECN1 gene, is a crucial autophagy promoter. This protein is a component of the PI3K complex, which regulates vesicle trafficking [136]. Beclin-1 deletion increases the chance of developing cancer in humans, while its deficiency has been linked to the onset of solid tumors. Mice heterozygous for BECN1 are predisposed to developing tumors for unknown reasons [137]. The mTOR complex 1 (mTORC1) regulates cellular metabolic activity. Since the phosphorylated ULK1 complex inhibits autophagic function, when energy needs are met, the mTORC1 phosphorylates ULK1 and ATG13, switching the metabolic behavior to anabolism [138]. In addition, autophagy regulates mitochondrial function in the face of adversity. PTEN-induced putative kinase 1 (PINK1) is activated when mitochondria become dysfunctional. For the autophagy machinery to recognize and degrade mitochondrial outer membrane proteins, PINK1 activates the E3 ligase parkin (PARK2). Reducing metabolic stress and reactive oxygen species (ROS) generation can be achieved through the selective removal of damaged mitochondria [139].

The protein kinase CaMKK2 is a member of the serine/threonine-specific protein kinase family responsible for activating the calcium/calmodulin-dependent protein kinase. CaMMKs play crucial roles in various metabolic pathways, including those that control glucose homeostasis, adipose tissue development, food intake, and inflammation. Since ARs are overactive in PCa, CaMKK2 is also overexpressed in this disease as one of the AR target genes [140]. Knocking down CAMKK2 has recently been shown to cause changes that reflect vesicle trafficking disturbance. Because of its impact on organelle integrity and membrane trafficking, CAMKK2 sustains cell proliferation, which is consistent with the role of autophagy in cancer development [141]. CaMKK2 knockdown tumors showed greater necrosis, particularly in regions with less energy support, which is consistent with the pro-survival role of CaMKK2-mediated autophagy. CAMKK2 helps cells endure a nutrient-deficient tumor microenvironment, which indicates that CAMKK2 is necessary for PCa growth in vivo [142].

Other autophagy-related genes involved in PCa carcinogenesis have been found upstream of mTOR pathway activation, including LKB1/AMPK. LKB1 expression, and thus AMPK activation, are lost with the progression of PCa from normal to neoplastic tissue [143]. AMPK activity cross-talks with the PI3K, mTOR, and mitogen-activated protein kinase (MAPK) pathways [144]. It was demonstrated that p38MAPK inhibition directly influences the survival of PCa cells. Patients with pancreatic cancer who lack LKB1 expression may respond well to ADT with p38MAPK inhibitors. On the other hand, patients with PCa with high LKB1 expression should be candidates for dual kinase inhibition targeting both p38MAPK and AMPK or for a treatment targeting the autophagic apparatus.

In addition, PCa reveals an important molecular link between autophagy and apoptosis. TRAIL, or tumor necrosis factor-related apoptosis-inducing ligand, is a cytokine that triggers cell death. By connecting to the death receptors (DR-4 and DR-5), TRAIL induces cancer cell death by assembling a death-inducing complex consisting of FADD (Fas Associated Via Death Domain) and caspase-8 [145]. Hence, autophagosome production and consequent accumulation of autophagosomal marker LC3B-II demonstrate that TRAIL acts synergistically to stimulate autophagy in PCa cells [146]. Recently, researchers have begun looking into how an autophagic genetic signature affects PCa susceptibility and prognosis. Based on the data from The Cancer Genome Atlas (TCGA) Prostate Adenocarcinoma (PRAD) dataset, Cheng et al. have uncovered differentially expressed autophagy-related genes (DEARGs). There were sixteen DEARGs found, seven of which were linked to improved overall survival in patients with PCa. Furthermore, the prognostic genes encoding the Niemann–Pick C1 protein (NPC1), B-cell lymphoma 2 (BCL2) interacting protein 3 (BNIP3), and tumor protein p53 were linked to specific clinical features. The elevated expression of NPC1 and BNIP3 was dramatically associated with advanced pathological T stages, while NPC1 overexpression was significantly related to higher ISUP grades [147]. More than twenty autophagy-related genes (ARGs) were shown to influence disease-free lifespan concerning T status, N status, and Gleason score in a separate investigation [148]. More than 230 ARGs were discovered by comparing DEARGs from the TCGA dataset with The Human Autophagy Database. The obtained results align with those seen in prior studies of biopsy tissue. The expression of the four essential autophagy proteins and their association with cancer aggressiveness has been studied (the microtubule-associated protein 1 light chain 3A–LC3A, LC3B, Beclin 1, p62, lactate dehydrogenase 5–LDH5). High levels of LDH5 and a low Gleason score were correlated with high levels of cytoplasmic LC3A, LC3B, and p62 expression (present in >50% of tumor cells per slice). The additional prostatic invasion was also strongly associated with Beclin-1 overexpression in a broad sense [149]. These results suggest that ARGs and phenotypical expression of autophagy may be useful prognostic indicators.

#### 3.3.5. Metabolic Crosstalk in PCa Microenvironment

The tumor microenvironment facilitates communication between cancer and stromal cells through the secretion of various cytokines and growth factors and alterations in the extracellular matrix. This re-regulation of crosstalk ensures the proliferation of cancer cells in a hostile environment and their ability to enter the extracellular matrix and spread. Active interactions between cancer cells, cancer-associated fibroblasts (CAFs), and tumor-associated macrophages (TAMs) have been observed. CAFs generate stem-like properties and epithelial–mesenchymal transition in PCa and drive metabolic reprogramming in a reciprocal fashion. The expression of glucose transporter 1 is higher in CAFs. Myofibroblasts, which are the result of CAF differentiation, produce lactate and pyruvate. MCT1 transports these metabolites into epithelial cancer cells, where they are integrated into the Krebs cycle. Lactate stimulates the sirtuin 1 (SIRT1)/PGC-1 axis in cancer cells, leading to increased mitochondrial bulk and activity. Recent research has found evidence of mitochondrial transfer between CAFs and cancer cells [150]. Due to this connection, cancer cells no longer rely on glucose for anabolic pathways and cell proliferation, but rather, they rely on lactate. Moreover, CAFs supply abundant glutamine via oncogenic Ras activation [151].

Another type of stromal cell known to communicate with PCa is adipocytes. Likely, the excess in dietary FA, modifications in the insulin–IGF-1 axis, and greater levels of pro-inflammatory cytokines associated with obesity and excessive visceral fat may increase the likelihood of PCa progression to metastatic illness [152]. Phenotypic changes in tumor-surrounding adipocytes have been described, including the downregulation of adipocyte markers and decreased lipid storage, as well as the overproduction of pro-inflammatory cytokines and extracellular matrix-related molecules [153]. The expression of fatty acid transporters CD36 and FATP5 is elevated in various PCa cell lines, together with FFA absorption from surrounding adipocytes. Furthermore, tumor cells are subject to FFA-induced oxidative stress via NADPH oxidase 5, which aids tumor cell invasion via activation of the hypoxia-inducible factor 1-alpha/matrix metalloproteinase 14 pathway. It has been found that advanced and high-grade PCa is linked to a greater thickness of adipose tissue surrounding the prostate (periprostatic adipose tissue, or PPAT). Biochemical recurrence-free survival following radical prostatectomy may be shorter in those whose cancers have contact with PPAT than in those whose malignancies did not invade beyond the prostate capsule [154].

### 3.4. Microbiome Influence

A recent study indicates that the microbiome has a role in the development and progression of PCa [27]. Typically, the microbiome is examined in relation to its location in the gut. Human-essential amino acids, vitamins, monosaccharides, and short-chain FAs, such as butyrate, acetate, propionate, and isopropionate, are all produced by the intestinal microbiome [155]. Diet is the most influential element on the gut microbiome [156]. In rodents [157] and humans [158], a high-fat diet decreases the abundance of Bacteroidetes and increases the abundance of Firmicutes and Proteobacteria, resulting in gut dysbiosis. This manner of developing gut dysbiosis compromises gut wall integrity and increases gut permeability, resulting in a “leaky gut” that releases gut metabolites or bacterial components into systemic circulation, hence inducing metabolic inflammation [159]. Through the stimulation of the NFB-IL6-STATAT axis, gut dysbiosis has been demonstrated to enhance PCa development and docetaxel resistance [160].

Nevertheless, a microbiome is also present in urinary and genital organs, such as the prostate, seminal vesicles, and urine bladder [161]. Multiple investigations have confirmed the presence of bacteria and viruses in prostate tissue, both normal and malignant [162]. Cohen R. et al. (2005) identified *Propionibacterium acnes* as the most prevalent bacteria related to inflammation of prostate tissue [163]. Cavarretta et al. (2017) identified a greater presence of *Streptococcus* spp. in tumor and peri-tumoral regions compared to non-tumorous regions when comparing the microbial makeup of different PCa locations [164]. Miyake et al. (2019) correlated the presence of *Mycoplasma genitalium* with PCa and high Gleason scores [165]. Likewise, a number of studies have investigated the connection between the urine microbiota and PCa. According to Yu et al. (2015), Bacteroidetes, Alphaproteobacteria, Firmicutes, Lachnospiraceae, Propionicimonas, and Sphingomonas and Ochrobactrum were more prevalent in PCa patients than in patients with benign prostatic hypertrophy [166]. In a study conducted by Alanee et al. (2018), PCa patients’ urine microbiota exhibited a higher abundance of *Veillonella*, *Streptococcus*, and *Bacteroides*, and a reduced abundance of *Faecalibacterium*, *Lactobacillus*, and *Acinetobacter* [167]. All of these studies suggest that persistent inflammation, in conjunction with prostatic and/or urinary tract infections, creates an inflammatory milieu that promotes the formation of PCa precursor lesions, hence promoting prostate tumorigenesis [168].

## 4. Resistance to Antiandrogens in PCa Therapy

Metastatic castration-resistant prostate cancer (mCRPCa) represents a major problem for treatment. When it was first histologically identified and described in 1853 by London surgeon John Adams, PCa was thought to be a rare disease. Unfortunately, progress in diagnostics, life expectancy extension, and so called the western lifestyle have debunked this assessment. Today, it is known that this cancer is at the top in terms of the number of cases in many regions of the world; for example, in the USA, it is the leading malignancy in men and the second most common cause of death due to cancer [169,170].

Parallel to the progress in diagnostics, the therapy of this disease was also developed. The first attempt to implant radium into tumor tissue was made in 1913, and in 1938 estrogen injections and surgical castration were begun. The ADT started in the early-1940s with the discovery that androgen inhibition could improve the outcome of PCa therapy (Charles Brenton Huggins, Nobel Prize in Physiology and Medicine, 1966), giving an adequate alternative to operative castration. However, it was to be another three decades before diethylstilbesterol was introduced into PCa therapy [171]. Subsequently, in the mid-seventies, the application of chemotherapy began, and in the early eighties, GnRH analogues were also introduced into PCa treatment (Andrew Schally, Nobel Prize 1977). Due to the ineffectiveness of first-line ADT, such as surgical orchiectomy with long-acting GnRH agonists or GnRH antagonists, other therapeutic options have been introduced in CRPCa: taxanes, sipuleucel-T, abiraterone acetate, enzalutamide, and radium-223, but with poor outcomes [172].

Recently, the use of PARP inhibitors such as olaparib (inhibits poly-ADP-ribose polymerase, an enzyme that repairs damaged tumor cells) has been attempted, with promising results. In 2021, the FDA approved olaparib (Lynparza^®®^) as the first PARP inhibitor intended to treat mCRPCa; a year later, the European Commission did the same. Until now, PARP inhibitors have been used in cases where mutations in the genes for BRCA1 and BRCA2 (i.e., PALB2) have been shown to render the DNA repair system ineffective. The theory of synthetic lethality was put forward by Theodosius Dobzhansky back in 1946. The essence of this theory is that the inhibition of PARP, as an enzyme that has a reparative role in DNA damage (and that helps tumor cells to survive damage caused by cytotoxic drugs), will be effective only if is associated with mutations of other repair proteins with a similar role (i.e., BRCA1 and 2). Inhibition of PARP is otherwise effective in the therapy of ischemia-reperfusion damage to organs, where excessive activation of reparative enzymes consumes all the ATP in the cell and leads to cell death by the type of necrosis, and the antitumor activity of PARP inhibitors was demonstrated in vitro more than fifteen years ago [173,174].

### 4.1. The Mechanism of Antiandrogens Action in PCa Treatment

AR belongs to nuclear receptors, i.e., transcription factors, similar to estrogen receptors, receptors for glucocorticoids and thyroid hormones. The AR gene is located in the q11-12 regions of the X chromosome. It consists of 8 exons that encode a 919 amino acids protein of approximately 110 kDa. The AR protein has four regions: the N-terminal, the DNA-binding region, the hinge region, and the ligand-binding region. In the N-terminal part there are 19–25 repetitive glutamine sequences (CAG code); the fewer there are, the greater is the transcriptional activity of the AR gene and the greater the PCa risk [175,176].

The most important AR endogenous ligands are testosterone and dihydrotestosterone. AR antagonists are also its ligands that should have a higher affinity for AR than testosterone and dihydrotestosterone.

The mechanisms of action of antiandrogens are different [177]:Inhibition of testosterone secretion;Inhibition of androgen action.

#### 4.1.1. Inhibition of Testosterone Secretion

The first mechanism is achieved indirectly, via the hypothalamus–pituitary–gonadal axis or by direct inhibition of testosterone secretion. Continuous administration of gonadorelin analogues (GnRH) effectively blocks testosterone secretion via the indirect pathway (leuprolide, goserelin, triptorelin, histrelin, and nafarelin, for example). These analogues inhibit the secretion of luteinizing hormone (LH) in the adenohypophysis, since permanent stimulation of GnRH receptors on the membrane of gonadotropic cells gradually leads to their downregulation (intermittent stimulation would cause the opposite). After the initial stimulation of LH secretion, within a week, the subsequent regulation of receptors in the adenohypophysis begins to produce results in the form of a decrease in LH secretion, so that the maximum effects on the decrease in testosterone secretion are achieved after 3–4 weeks of treatment.

Other drugs with the same target are GnRH antagonists. In contrast with the former drugs, the latter immediately decrease the secretion of LH without its temporary surge. In 2021, Liu et al. reported that hundreds of GnRH antagonists were synthesized in the last fifty years [178]. They were discovered in 1972, and two decades later, cetrorelix, a GnRH antagonist with an improved pharmacological profile, was the first to be clinically tested on patients with PCa. However, neither cetrorelix nor the first clinically used GnRH antagonist abarelix were free of histamine-related serious adverse reactions, such as oedema-promoting effects and allergic reactions. In 2008, the next-generation GnRH blocker with an improved safety profile, degarelix, was introduced into clinical practice worldwide [179]. Finally, relugolix (Orgovix^®®^), a new GnRH blocker for oral use, received FDA approval in 2020.

Another mechanism of inhibiting the secretion of testosterone is the blockade of the testosterone synthesis enzyme. For example, ketoconazole blocks cytochromes (CYP enzymes), but azoles are not used for pharmacotherapeutic purposes as antiandrogens because of side effects at higher doses (adrenal insufficiency and hepatotoxicity). However, a newer drug blocks 17α-hydroxylase and C-17,20-lyase (CYP17A1) in testicular, adrenal, and PCa cells, and has considerable efficacy in PCa therapy. It is abiraterone, which reduces the conversion of pregnenolone and progesterone into their 17α-hydroxy derivatives, thereby reducing the formation of DHEA and androstenedione. Thus, circulating testosterone levels become almost unmeasurable. Abiraterone also blocks the AR to some extent, as well as the synthesis of other steroids (Figure 1) [180].

Finally, abiraterone is an androgen biosynthesis inhibitor in testicular, adrenal, and prostatic tumor tissue targeting CYP17. It is also an AR antagonist and 3β-hydroxysteroid dehydrogenase inhibitor with affinity for CYP11B1 and a panel of hepatic CYP enzymes. A newer drug, galeterone, also blocks CYP17, AR, and the transcriptional activity of this receptor in the nucleus of tumor cells [181].

#### 4.1.2. Inhibition of Androgen Action

Other mechanisms of therapeutic targeting of the AR pathway involve pharmacological antagonism at the receptor level and inhibition of the conversion of testosterone into a more active metabolite—DHEA—which is achieved by blocking the enzyme 5α-reductase. Examples of AR antagonists include: flutamide, bicalutamide, nilutamide, apalutamide and enzalutamide (Table 1).

These antiandrogens bind to the ligand binding site on the AR and thus prevent androgen binding. Unlike bicalutamide, apalutamide is an antiandrogen with a high affinity for the AR that does not have agonistic activity (thus, it is not a partial agonist, but a true antagonist); it also blocks the binding of AR to DNA in the nucleus of PCa cells. Darolutamide has an advantage over enzalutamide and apalutamide, as it blocks both overexpressed and mutated AR, which is the main mechanism of resistance to the other two mentioned antagonists. Additionally, darolutamide blocks the transcriptional activity of AR with point mutations F877L, H875Y/T878A, F877L/T878A, and T878G. The clinical efficacy of AR antagonists is lower when they are used as monotherapy, since AR blockade leads to the termination of the negative feedback loop and increased secretion of LH and thus of testosterone. Better results in the treatment of metastatic PCa are achieved if AR antagonists are combined with GnRH analogues; then AR antagonists block the effects of adrenal androgens that GnRH analogues do not. A newer approach in antagonizing the AR is the blockade of the AF-1 region (activation function-1) on the N-terminal part of the AR and this is the mechanism of action of the new substance, EPI-506, which is the precursor of the active molecule EPI-002; said antagonist is in phase I/II clinical trials for patients with mCRPCa resistant to abiraterone and/or enzalutamide. Spironolactone is also a weak antagonist of AR and testosterone biosynthesis, and cyproterone acetate, a progestin, can block AR to some extent. On the other hand, finasteride and dutasteride, as 5α-reductase inhibitors, prevent testosterone from converting into DHEA, especially in the male external genitalia. Both drugs are used in the treatment of benign prostatic hypertrophy.

### 4.2. Resistance to Antiandrogens

It is estimated that 20–40% of PCa patients have a recurrence after initial radiation therapy or surgical removal of the prostate. Then, drugs that suppress endogenous androgens (androgen deprivation therapy, ADT) are introduced into the treatment. However, 12–18 months after initiation of ADT, CRPCa develops with disease progression. AR gene amplification has been identified in 30–50% of CRPCa cases and is associated with poor prognosis and shorter overall survival, as well as shorter progression-free survival [170].

One of the important causes of resistance to antiandrogens in PCa therapy is changes at the gene level. Muller et al. (2012) showed ten years ago that in some tumor tissues there is a co-deletion of genes that suppress tumor development and they are located in the same part of the chromosome [182]. Such is the case with the T53 gene. Similarly, co-amplification of nearby tumor-stimulating genes has been demonstrated. For example, OPHN1 (oligophrenin 1 gene) is located in the Xq12 region, as is the AR gene. OPHN1 protein is activated under the influence of Rho GTPase (Rho-GTPase-activating protein, Rho-GAP). Otherwise, the Rho family of GTPases includes small signaling G proteins that regulate gene expression by acting on the dynamics of actin, a protein that plays a key role in transcription, chromatin remodeling, and primary mRNA transcript processing [183]. Non-functional (aberrant) expression of the OPHN1 gene is associated with various types of mutations (incorporation of wrong nucleotides—missense; wrong incorporation of a stop codon that prematurely stops protein synthesis—nonsense; insertion or deletion of one or more bases that leads to misreading of DNA—frameshift) and chromosomal deletions.

Overall, antiandrogen therapy suppresses most PCas for a certain period of time, but resistance to ADT is virtually inevitable. Nearly all cancers will progress to CRPCA if given ADT for an adequate period of time. The resistance mechanisms leading to CRPCA can be divided into five groups:Point mutations of the AR gene;Amplification of AR and CYP17;Changes in androgen biosynthesis;Changes in cofactors for AR;Creation of different variants of AR.

#### 4.2.1. Point Mutations of the AR Gene

These mutations are rare in primary hormone-dependent tumours but occur more frequently in CRPCa (>10%), especially in cases where the tumor has progressed despite systemic hormone therapy and the use of new antiandrogens. Not only that, but AR gene mutation is among the nine most significant gene mutations in CRPCa [184]. The result of these mutations varies, from loss of receptor function to gain or loss of associated intracellular signaling pathways. The changes are located mainly in the part of the receptor to which the ligand binds (LBD, ligand-binding domain). It is the non-specific activation of the LBD receptor for androgens as a result of point mutation that can potentiate the response of the receptor to other ligands, which stimulates tumor growth. The T878A mutation occurs during therapy with abiraterone, which inhibits CYP17, due to which the synthesis of dehydroepiandrosterone and testosterone is suppressed (see Figure 1) and progestin production is increased. The significance of this becomes greater considering that this mutation increases the sensitivity of AR to steroid hormones such as estrogens and progestins, but also to antiandrogens, which then become full AR agonists [185]. It is interesting that the T878A mutation was identified in almost a fifth of the mCRPCa samples that progressed on abiraterone, but also that the loss of efficacy of one antagonist (i.e., flutamide) does not mean a loss of another (i.e., bicalutamide). Another important mutation is F877L, which occurs during the administration of the newer antagonists, enzalutamide and apalutamide, and which, for example, convert enzalutamide from an antagonist to a full AR agonist. This mutation alters the LBD of the AR but does not alter the antagonistic activity of bicalutamide. Additionally, the L702H mutation occurs in patients treated with a combination of abiraterone and glucocorticoids (dexamethasone or prednisolone) and leads to a refractory form of cancer to further abiraterone therapy. In this mutation, enzalutamide has no inhibitory effect and dexamethasone acts as an activator.

#### 4.2.2. Amplification of Genes for AR and CYP17

This resistance mechanism often occurs with ADT and plays an important role in the loss of efficacy of the new antiandrogens, enzalutamide and abiraterone. It has been identified in as many as 80% of CRPCa cases [186]. At the same time, there is a threefold increased expression of both the complete AR in its full length, as well as altered variants (splice—interlaced forms with a shortening at the C-terminal part that do not have the classic LBD). During abiraterone therapy, there is a two-fold increase in CYP17A1 gene expression in relapsed forms of CRPCa. A higher copy number of free circulating AR and CYP17A1 indicates a worse prognosis in those treated with abiraterone [187].

#### 4.2.3. Changes in Androgen Biosynthesis

Tumor cells can escape inhibition of androgen synthesis by goserelin or abiraterone by increasing, intratumorally, androgen synthesis and/or AR, but also via glucocorticoid receptors and glucocorticoids [188]. Evading the effect of the drug or creating new metabolic pathways of salvation are not only important mechanisms of tumor defense against the effects of drugs but are also grounded in the very biological basis of the organism as a defense of reproductive capacity, i.e., fertility independent of tumor development [189].

In conditions of AR blockade due to ADT, rescue pathways are activated in the form of at least three alternative signaling pathways that involve growth factors and stimulate AR signaling pathways independently of androgens, i.e., steroids (Table 2):The EGF/IGF signaling pathway that is mediated by phosphatidylinositol 3-kinase (PI3K) and protein kinase B (Akt/PKB) and depends on the level of phosphatidylinositol 3,4,5-triphosphate (PIP3) and PTEN in the cell;The Ras signaling pathway (Ras family of proto-oncogenes) that occurs via Cdc42-linked kinase (Ack), Ras/Raf mitogen-activated protein kinase/ERK kinase (MEK), and the proto-oncogene tyrosine protein kinase Src;IL6 cytokine signaling pathway that activates AR through Janus kinase signal transducer and activator of transcription (JAK1), signal transducer and activator of transcription 3 (STAT3), and intermediate histone acetyltransferasep300.

In a tumor consisting of several million cells, there are already those that have activated drug resistance mechanisms and will contribute to the development of CRPCA. Induced resistance is also possible. However, tumor size does not necessarily increase the likelihood of developing CRPCA. There are many other factors (i.e., presence of aggressive variant PCa alterations). Shore et al. (2021) emphasized predictive factors for progression to CRPCA: Gleason score, extent of metastatic spread, hereditary genetic factors/mutations, and hormonal kinetics [190]. For example, risk of CRPCA is lower if PSA rises more slowly during the first off-treatment interval in patients on intermittent ADT. Additionally, a set of biomarkers has been recently established as prognostic tools in CRPCA progression: lactate dehydrogenase, Eastern Cooperative Oncology Group performance status, presence of liver metastasis, serum albumin, alkaline phosphatase, and time from start of initial ADT to start of abiraterone treatment. 

#### 4.2.4. Changes in Cofactors for AR

Signal transduction of all nuclear receptors, including AR, begins with cofactor recruitment. Previously, the ligands bound to the receptors. They subsequently move into the nucleus and form dimers with a conformational change into an active form that can interact with promoter regions. The dimer–cofactor complex (coactivators or corepressors) activates or inhibits transcription. The ligand nature determines the receptor conformation and cofactor affinity [191,192].

Since 1995, when the first cofactor, SRC-1, was isolated, their number has increased to over 200 [193]. They can be divided into four groups: (i) molecular chaperones that coordinate AR maturation and movement, (ii) histone modifiers (i.e., CBP/p300, NCoR), (iii) coordinators of transcription (i.e., TRAP/DRIP/ARC), and (iv) DNA structural modifiers (i.e., SWI/SNF/BRG1) [194].

Li et al. (2002) showed that changes in the expression of AR cofactors could play a vital role in the development of prostate tumors. They confirmed by in situ hybridization that in PCa epithelial samples, there is increased expression of Ran/ARA24 and PIAS1 and slightly less of SRC1, TMF1/ARA160, TRAP220, ARA54, and ELE1/ARA70 [192]. The authors considered only eight of over 200 cofactors identified so far.

Fujita et al. (2019) showed that the expression of proteins from the SRC family (steroid receptor coactivator), SRC1, -2, and -3, is correlated with the aggressiveness of PCa [176]. In mice, SRC-3 is essential for the progression and dissemination of this cancer, but SRC1 is not. On the other hand, androgen deprivation increases the expression of SRC2, further stimulating PI3K signaling, promoting PCa metastasis, and leading to the development of CRPCA. The effect of SRC2 on lipid metabolism seems crucial. Namely, increased carboxylation of alpha-ketoglutarate leads to transcriptional reprogramming of glutamine metabolism, which resets the tumor metabolism to support its uncontrolled growth and survival. Finally, Tien et al. (2013) showed that SRC3 levels are increased in human CRPCA and PTEN-negative PCa and reduced patients’ recurrence-free survival. In CRPCA, SRC3 stimulates the expression of Akt and S6 kinase B1 (S6K1) [195]. Of note, the oncostatic speckle-type pox virus and zinc finger (POZ) protein (SPOP) mutation may also be involved.

Another AR coactivator, CBP/p300, also has an oncogenic effect. It increases during androgen deprivation, and blocking its bromodomain reduces tumor growth [196]. Similarly, there are oncogenic coactivators, such as Tip60 (increases AR translocation into the nucleus), four-and-a-half LIM protein 2 (increased expression of FHL2 is observed in cancer recurrence), and Hic-5/ARA55 (involved is in the androgenic activation of keratinocyte growth factor, a paracrine stromal protein).

#### 4.2.5. Creation of Different Variants of AR

Wild-type full-length AR (AR-FL) consists of an N-terminal domain, a DNA-binding domain, and a ligand-binding domain (NTD, DBD, and LBD, respectively), encoded by exon 1, exons 2–3, and exons 4–8, respectively. Along with AR-FL, in CRPCA, splice variants are also particularly prevalent: AR-V1, AR-V7, and AR-567. Splice variants have a low abundant LBD and can be constitutively active in androgen-responsive genes even without ligands. At the same time, AR-FL interacts with both coactivators and corepressors, while AR-V7 is mainly linked to corepressors. This is partly due to isoform-specific variations in H3K27 acetylation (post-translational modification with the epigenetic effect of lysine in histone H3; namely, AR-V7 increases the expression of histone deacetylases through corepressors, which decreases H3K27 acetylation). The role of the HOXB13 gene and its product, the transcription factor, is vital for AR-V7 signaling as well.

CRPCA resistance to enzalutamide results from the absence of the target (i.e., LBD), whereas abiraterone increases the expression of ARf1, AR-V7, and AR-567 [197,198]. AR-V7 has been the best studied so far due to the discovery of a specific antibody for immunohistochemical tissue staining [199]. In a 2011 study, AR-V7 was detected in 100% of CRPCA bone metastases samples (30/30), as was AR-V1, while AR-V567 was only detected in 23% of cases (7/30 samples); the prevalence of these splice variants was significantly lower in hormone-naive samples (80%, 90%, and 0%, respectively) [200]. In enzalutamide-treated patients with CRPCA, AR-V7 was detected in 39% of cases and in abiraterone-treated patients in 19% of cases [201].

## 5. Treatment of Metastatic Prostate Cancer

From Charles Huggins and androgen deprivation in the treatment of mPCa [202] to the next study that prolonged the overall survival (OS) of TAX-327 [203], several decades passed. In the last ten years, the treatment of mPCa has been developing rapidly and almost every year we receive a new standard for the treatment of this disease. Treatment of mPCa can be roughly divided into three phases: treatment of metastatic castration-sensitive prostate cancer (mCSPCa), mCRPCa treatment, and, specific to this disease, castration-resistant PCa without visible metastases by conventional imaging (M0 CRPCa) [204]. Although it is a later stage in the treatment of mPCa, innovative drugs were first investigated in the treatment of mCRPCa, so we will focus this review on this stage of the disease.

### 5.1. Treatment of Metastatic Castration-Sensitive Prostate Cancer

Until just a few years ago, the mCSPCa treatment was exclusively via androgen deprivation with either luteinizing hormone-releasing hormone (LHRH) analogues, LHRH antagonists, or subcapsular orchiectomy. Attempts to add peripheral anti-androgens of the first generation did not lead to a prolongation of patient survival, so their administration was not recommended by experts [205]. Today, practically no patients are treated with ADT alone.

The first study to add a specific therapy to ADT was the CHAARTED study, which compared docetaxel with placebo in mCSPCa patients. Patients were stratified into a group with “high volume” disease, which is defined by the presence of visceral metastases and/or at least four bone metastases, at least one of which is in the peduncle of the skeleton; and another group that does not have these characteristics called “low volume”. Although the study demonstrated longer survival in the intent-to-treat (ITT) population, a sub analysis found that only those patients with high-volume disease benefited (OS 57.6 vs. 44 mos; HR 0.61 95% CI 0.47–0.8; *p* < 0.001) [206]. Given these results, docetaxel is reserved for those patients with high-volume disease who are fit enough to receive chemotherapy.

LATITUDE is a study that examined the addition of abiraterone to ADT in patients with initially metastatic castration-sensitive “high-risk” PCa, defined by the presence of at least two of three features: visceral metastases, at least three bony metastases, and a Gleason score ≥8. In this study, 1199 patients were randomized to either abiraterone and prednisolone or placebo and prednisolone. All patients were on ADT. Co-primary endpoints were overall survival and radiological time to disease progression. Abiraterone prolonged OS compared to placebo (not reached vs. 34.7; HR 0.62 95% CI 0.51–0.76; *p* < 0.001) and rPFS (33.0 vs. 14.8 mos; HR 0.47 95% CI 0.39–0.55, *p* < 0.001). Grade 3 side effects of hypertension and hypokalemia were more common in the abiraterone group [207].

Apalutamide is a newer generation antiandrogen. In the study, TITAN was compared with placebo in patients with mCSPCa. The study included all patients regardless of volume or risk of disease and all were on ADT. Primary endpoints were rPFS and OS. In the primary analysis, after a median follow-up of 22.7 months, the proportion of patients who were free of radiological progression in the apalutamide group at two years was 68.2% versus 47.5% in the placebo group (HR 0.48 95% CI 0.39–0.60, *p* < 0.001). The most common side effect of apalutamide was rash [208]. Although the primary analysis did not reach statistical significance, the update analysis showed an OS benefit of apalutamide (median OS not reached v 52.2 mos; HR 0.65; 95% CI 0.53–0.79; *p* < 0.0001) despite crossover that was allowed in the study [209]. Apalutamide has shown benefit regardless of volume and level of disease risk [208,209].

Similar to apalutamide, enzalutamide has also demonstrated efficacy in mCSPCa. The ARCHES study compared enzalutamide with placebo in addition to standard ADT. Patients were stratified by disease volume and prior docetaxel use. In all subgroups, enzalutamide prolonged the time to radiological disease progression. Adverse effects of Grade 3 and greater were equally common in both groups [210]. In an updated analysis, enzalutamide prolonged overall survival compared to placebo (HR 0.66 95% CI 0.53–0.81, *p* < 0.001) [211]. The ENZAMET study, similar to the previous one, aimed to prove the efficacy of enzalutamide in mCSPCa, but in combination with bicalutamide. Also in this study, enzalutamide demonstrated longer rPFS and OS regardless of disease volume and prior docetaxel use. The most common side effect of enzalutamide was fatigue, while convulsions occurred in seven patients in the enzalutamide-treated group [212].

The most recent treatment concept for mCSPCa is the so-called triple therapy, which includes simultaneous administration of ADT, antiandrogens, and docetaxel chemotherapy. The PEACE-1 study is a 2 × 2 factorial design study in which a control group of patients received standard of care (SOC) ADT plus docetaxel, one study group received SOC plus radiotherapy, one group received SOC plus abiraterone, and the last group received SOC plus abiraterone and radiotherapy. A total of 1173 patients were randomized in the study. Patients receiving abiraterone had longer rPFS (HR 0.54 99.9% CI 0.41–0.71, *p* < 0.0001) and OS (HR 0.82 95.1% CI 0.69–0.98, *p* = 0.03). The addition of abiraterone to ADT and docetaxel did not increase the rates of neutropenia, febrile neutropenia, fatigue, or neuropathy. Triple therapy achieved the greatest benefit in high-volume disease according to CHAARTED criteria [206,213]. Another study that examined the effect was the ARASENS study, which examined the addition of the antiandrogen darolutamide to the standard combination of docetaxel and ADT. Darolutamide demonstrated longer rPFS and OS compared to the non-darolutamide group. The most common adverse effect was neutropenia, and the rate did not differ between the darolutamide and placebo groups [214].

### 5.2. Treatment of Metastatic Castration-Resistant Prostate Cancer

As already mentioned, for a long time ADT was the only option that prolonged the survival of mCRPCa. The TAX-327 study examined three chemotherapy options in mCRPCa, three-weekly docetaxel, weekly docetaxel, and mitoxantrone. Symptomatic patients with mCRPCa, as well as those who had visceral metastases, were included in the study. The primary target was OS. Both docetaxel groups had longer OS compared to mitoxantrone. Since 2004, three-week docetaxel has become the standard treatment for symptomatic patients with mCRPCa [203]. A little later, studies began to be conducted after progression to docetaxel treatment. Study COU-AA-301 examined abiraterone acetate (AA) plus prednisolone versus prednisolone plus placebo in patients with mCRPCa who had previously received chemotherapy. In the study, 1195 patients were randomized at a ratio of 2:1 in the branch with AA. The primary target was OS. Abiraterone acetate demonstrated longer OS compared to placebo (14.9 vs. 10.9 mos; HR 0.65 95% CI 0.54–0.77, *p* < 0.001). In addition, all key secondary objectives were achieved in this study: time to PSA progression, radiological time to disease progression (rPFS), as well as PSA response rate. Fluid retention, hypertension, and hypokalemia were the most common side effects in the AA group [215]. Enzalutamide, a newer generation anti-androgen, was investigated in the AFFIRM study, versus placebo, after progression to chemotherapy. In the ratio 2:1, 1199 patients were randomized to receive enzalutamide or placebo. The primary endpoint of the study, OS, was longer in the enzalutamide group (18.4 vs. 13.6 mos; HR 0.63 95% CI 0.53–0.75, *p* < 0.001). Similar to AA, all secondary objectives were met in the study with enzalutamide. The most common side effects in the enzalutamide group were hot flashes, fatigue, and diarrhea, while 0.5% of patients had seizures [216].

A large number of patients enter the mCRPCa stage of the disease due to a jump in PSA or the appearance of new asymptomatic metastases. Considering this fact, studies were developed for mCRPCa in asymptomatic or minimally symptomatic patients who had not previously received docetaxel. Study COU-AA-302 compared AA with placebo in combination with prednisolone. The study enrolled 1088 patients and the co-primary endpoints were rPFS and OS. Median rPFS was longer in the AA group (16.5 vs. 8.3 mos; HR 0.53, 95% CI 0.45–0.62, *p* < 0.001), as was median OS (NR vs. 27.2 mos; HR 0.75 95% CI 0.61–0.93; *p* = 0.01) [217]. In a study of similar design, PREVAIL, enzalutamide and placebo were compared in docetaxel naïve patients. In total, 1717 patients were included in the study. Enzalutamide demonstrated longer 1-year rPFS (65% vs. 14%, HR 0.19 95% CI 0.15–0.23, *p* < 0.001) and longer OS at the time of publication. In the enzalutamide group, 72% of patients were alive, while in the placebo group, that percentage was 63% (HR 0.71 95% CI 060–0.84, *p* < 0.001) [218]. In addition to the aforementioned phase III studies, enzalutamide was compared with bicalutamide, an anti-androgen of the older generation, in the phase II TERRAIN study. And in this study, enzalutamide demonstrated superior rPFS (15.7 vs. 5.8 mos, HR 0.44 95% CI 0.34–0.57, *p* < 0.0001) [219]. The safety profile of AA and enzalutamide was similar to that seen in the post-docetaxel phase studies [215,216].

Cabazitaxel is a taxane cytostatic. In the TROPIC study, it demonstrated longer PFS and OS compared to mitoxantrone when discontinued after progression to docetaxel therapy [220]. After the results of the TROPIC study, as well as the results with docetaxel, AA, and enzalutamide, the question of the optimal drug sequence in the treatment of mCRPCa was raised. A partial answer to this question was given by the CARD study [221]. The CARD study compared cabazitaxel versus AA or enzalutamide in patients who had previously progressed on docetaxel and one of the ART drugs. Patients receiving enzalutamide were randomized to AA and vice versa. A total of 255 patients were randomized. The primary objective was rPFSm, the secondary objectives were OS and safety. After a median follow-up of 9.2 months, 73.6% of patients in the cabazitaxel group, versus 80.2% of patients on ART, had radiologic progression or died (HR 0.54 95% CI 0.40–0.73, *p* < 0.001). Median OS was longer in the cabazitaxel group (13.6 vs. 11.0 mos; HR 0.64, 95% CI 0.46–0.89, *p* = 0.008). The most common side effect of Gr3 cabazitaxel was neutropenia in 44.7% of patients.

Radiopharmaceuticals are an interesting field of mCRPCa treatment. The first registered radiopharmaceutical for the treatment of this disease was the alpha emitter Radium-223, which demonstrated longer OS compared to placebo in docetaxel-naïve patients with mCRPCa. This type of treatment did not show significant side effects but was reserved only for patients with bone metastases [222]. The VISION study is a phase III study that compared the radiopharmaceutical Lutetium-177-PSMA-617 with investigator’s choice of standard therapy in pre-treated patients with mCRPCa who had previously received at least one ARTA drug and one or two taxanes. 177Lu-PSMA617 significantly prolonged rPFS (8.7 vs. 3.4 mos; HR 0.40 99.2% CI 0.29–0.57, *p* < 0.001) and overall survival (15.3 vs. 11.3 mos, HR 0.62 95% CI 0.52–0.74, *p* < 0.001). Grade 3 side effects were more frequent in the 177Lu-PSMA617 group, but quality of life was preserved [223].

Comprehensive genomic profiling (CGP) and personalized therapy of malignant diseases is the focus of interest of the oncology public. ESMO recommends CGP in the treatment of mCRPCa to select the most adequate therapy [224]. Several biomarkers have shown significance in the treatment of mCRPCa. ProFOUND is a study that looked at olaparib compared to ART in patients who had previously been treated with another ART. The study had two cohorts in relation to biomarkers expressed on tumor tissue. Cohort A included somatic *BRCA1*, *BRCA2,* and *ATM* mutations, while cohort B included patients with 12 other prespecified genes. Cohort A included 245 patients, while cohort B included 142. Olaparib prolonged PFS in cohort A (7.4 vs. 3.6 mos; HR 0.34 95% CI 0.25–0.47, *p* < 0.001). In addition, benefit was shown in objective response (ORR) and time to PSA progression. Additionally, rPFS was prolonged with olaparib in the entire study population (cohorts A and B). The most common side effects of olaparib were anemia and nausea. Statistical significance for overall survival has not yet been proven in the aforementioned analysis [225]. However, already in the next interim analysis, olaparib prolonged OS in cohort A [226]. In the recently published ProPEL study, olaparib was investigated as an add-on to abiraterone in the first-line treatment of mCRPCa. Patients were stratified according to the presence of homologous recombination repair mutations (HRRm). Interestingly, olaparib prolonged PFS in both cohorts regardless of the presence or absence of HRRm [227]. We are waiting for regulatory bodies to publish guidance regarding the need for HRR testing in patients we plan to treat with olaparib.

Immunotherapy with checkpoint inhibitors (ICI) plays a significant role in the treatment of various urological tumors [228,229]. In PCa, ICIs do not yet have such a prominent role. Negative studies with the CTLA-4 antibody ipilimumab [230] and the addition of the PD-L1 antibody atezolizumab to enzalutamide in the treatment of mCRPCa [231] are the only completed phase III studies with ICIs. However, ICI therapy with pembrolizumab was included in the NCCN guidelines for the PCa treatment [232], based on a tumor agnostic study that included only patients with mismatch–repair deficient (MMR-D) tumors [233]. MMR-D occurs in about 5–10% of prostate tumors [234].

### 5.3. Treatment of Non-Metastatic Castration-Resistant Prostate Cancer

The definition of castration resistance in addition to radiological progression includes an increase in PSA with the patient’s castrate testosterone level. Sometimes PSA progression occurs without visible metastases by conventional imaging methods including skeletal scintigraphy and CT scans. What is also the fact that in this situation, patients with a shorter PSA doubling have a shorter time to the appearance of metastases and a worse prognosis [235]. In 2015, the St. Gallen consensus determined an arbitrary PSA doubling time of less than 10 months as a cut-off for the design of clinical studies in M0 CRPCa [205]. Additionally, with the application of newer imaging methods such as PSMA-PETCT, there are fewer and fewer patients with classic M0 CRPCa because metastases are found in most patients with high risk M0 CRPCa if this imaging method is applied [236]. Three different newer generation antiandrogens have been investigated in the treatment of patients with M0 CRPCa. The PROSPER (enzalutamide), ARAMIS (darolutamide), and SPARTAN (apalutamide) studies all showed significant results in delaying the occurrence of metastases and prolonging the lives of patients, with the metastasis-free survival as their primary objective [237,238,239]. As a result, early intervention with antiandrogen drugs in high risk M0 CRPCa became the standard of treatment [204,232].

### 5.4. Treatment of Castration-Resistant Prostate Cancer and Autophagy

Autophagy is a survival mechanism that cancer cells use to deal with metabolic stress caused by various treatments, such as castration therapy. Additionally, autophagy has been shown to promote castration resistance. Evidence suggests that the expression of CAMKK2, AMPK, and ULK1 in males with PCa is related to the disease’s prognosis and progression. When AMPK-ULK1 signaling is inhibited, autophagy, cell proliferation, and colony formation are also inhibited in Lymph Node Carcinoma of the Prostate (LNCaP) cells. These findings add to the growing evidence that the CAMKK2-AMPK-ULK1 signaling cascade promotes autophagy and, thus, PCa and castration resistance [240].

Despite their distinct action methods, abiraterone and enzalutamide are currently used in treating CRPCa. Enzalutamide is a drug that exerts multiple effects on androgen signaling, including: (i) the inhibition of testosterone binding to AR, (ii) the suppression of AR nuclear translocation, and (iii) the inhibition of AR binding to DNA. It has been demonstrated that the administration of enzalutamide induces an autophagy cascade in the C4-2B subline of LNCaP cells, which is mediated by the activation of AMPK and the inhibition of mTOR. This subpopulation of cells can replicate the behavior of CRPCa in humans by growing in castrated hosts. Moreover, it has been demonstrated that inhibiting AMPK in cells exposed to enzalutamide inhibits autophagy and promotes cell death. These results indicate that autophagy is a crucial survival strategy in CRPC [241]. However, Abiraterone and enzalutamide’s effects on cells go beyond the simple inhibition of AR activity: both have been shown to inhibit PCa proliferation and promote apoptosis via mitophagy regulation. The role of mitophagy in drug action is well established: it has been demonstrated that the addition of a mitophagy inhibitor, mitochondrial division inhibitor 1, can increase proliferation while limiting apoptosis in PCa cells [242]. However, inhibiting autophagy has been shown to improve the efficacy of abiraterone. Mortezavi et al. (2019) found that abiraterone increased the levels of ATG5 and LC3II in PCa cells, highlighting the upregulation of autophagy [243]. They also demonstrated that autophagy inhibitors could significantly reduce cell viability, increasing apoptosis. They also demonstrated that autophagy inhibitors could significantly reduce cell viability, increasing apoptosis.

The same evidence has been presented in the study of apalutamide, another AR antagonist used in CRPCa. Apalutamide treatment of LNCaP cells induces autophagy, which has been shown to have a strong anti-tumor effect when combined with autophagy inhibitors such as 3-methyladenine or siRNAs [244]. LNCaP cells express highly sensitive AR in their cytosol, making them highly androgen-sensitive. The AR of LNCaP cells contains a single point mutation in the ligand-binding domain that changes the sense of codon 868 (Thr to Ala) [245]. Nonetheless, the AR cannot fully explain autophagy regulation. Under ADT, PCa cell lines lacking androgen receptors, for example, can increase autophagic flux. This could be explained by disruptions in other proto-oncogenes, such as PTEN, or mutations in tumor suppressors, such as p53, which are associated with aberrant expression of downstream AR targets [246].

Standard treatment for men with metastatic PCa or CRPCa includes the microtubule-stabilizing drug docetaxel. A number of studies have demonstrated that autophagic activity during taxane-based therapy varies considerably. While some research suggests that taxane therapy may stimulate autophagy, others have found the opposite to be true [246,247]. However, it is well established that the autophagic response to taxanes is context dependent, depending on the duration of therapy and on environmental factors that contribute to the degree of autophagic activity [247]. Despite the apparent controversy surrounding the data, they supports the idea that autophagy can be used in various contexts. It is common knowledge that autophagy is essential in this context for taxane resistance. As recently confirmed, docetaxel-resistant melanoma cells can increase autophagic flux by increasing Forkhead box protein M1 (FOXM1) expression. Additionally, a drop in autophagic flux and a reduction in the number of autophagosomes were observed after FOXM1 was knocked down, making these cells more sensitive to docetaxel both in vitro and in vivo [248].

Clinical responses after receiving prostate-specific membrane antigen (PSMA)-targeted radioligand treatment (RLT) with 177Lu or 225Ac radionuclides have been seen in between 57 and 76% of PCa patients [249,250]. However, a lack of understanding of the underlying processes of resistance is a major obstacle to creating more effective RLT, underlining the potentially significant role of autophagy. Recent research indicates that the activation of ATG5-mediated autophagy in response to a lack of glutamine is a tumor survival strategy that allows cells to endure radiation-induced cell damage. It has recently been demonstrated that activating ATG5-mediated autophagy in response to a glutamine deficiency is a PCa survival strategy for surviving radiation-induced cell damage [251]. Furthermore, activation of the PI3k/mTOR pathway has been linked to the development of radio-resistance, and several studies have been conducted to investigate the radio-sensitizing effects of mTOR inhibitors. However, using mTOR inhibitors as radio sensitizers for RLT remains controversial, as preclinical studies have yielded contradictory results [252].

## 6. Conclusions

ADT remains the mainstay of primary PCa pharmacotherapy, but not as a monotherapy in most metastatic patients. Remission after ADT usually lasts 1–4 years, although in some cases it is longer (10 years). New strategies in the treatment of this malignancy include the development of inhibitors of androgen signals, the use of a combination of antiandrogens as a gold standard, as well as intermittent regimens of these drugs. Survival is prolonged, but still the cancer returns; that is, there is an exacerbation. It is a fact that even in CRPCa, cancer cells are dependent on androgens, but the mechanism of resistance to antiandrogens often remains unknown [189,253,254].

Not all targets listed in Table 2 are suitable for drug development (druggable) or occur in cell cultures in vitro but not in vivo. Additionally, combinations of antiandrogens (i.e., abiraterone + enzalutamide) have not been shown to be sufficiently effective in clinical trials [255]. The role of stem cells in tumor tissue is also controversial, regarding whether it is a reservoir for the development of resistance to antiandrogens. Potential new drugs that may help in the therapy of mCRPCa are PARP inhibitors. The FDA recently registered olaparib and rucaparib for this indication. Various studies have shown that genes for proteins involved in DNA repair are altered in 11.8–27% of mCRPCa cases [256]. Phase II and III clinical studies are ongoing examining the efficacy and safety of olaparib, rucaparib, talazoparib, and niraparib in patients with mCRPCa, either as monotherapy or in combination with other drugs (i.e., abiraterone, enzalutamide, carboplatin, docetaxel, durvalumab, or prednisone).

Another novel approach in PCa treatment could be modulation of autophagy. There have been a number of attempts to restore sensitivity to docetaxel or ADT by inhibiting autophagy. In preclinical models, the results have been encouraging because blocking autophagy has been shown to increase cell death in response to anti-cancer treatment. Currently, the focus of research is on conducting clinical trials to verify the findings of preclinical reports in patients with refractory malignancies [246]. Small interfering RNAs (siRNAs) and 3 methyladenine may be effective autophagy inhibitors in vitro. However, siRNAs are not routinely used in clinical practice and have been confined to confirmatory tools in preclinical research. Alternately, chloroquine and its derivative, hydroxychloroquine, may be more clinically relevant autophagy inhibitors [257]. The potential benefits of using chloroquine to treat patients are being tested in ongoing clinical trials (NCT04011410, NCT00726596).

Recently, the efficacy of chloroquine in combination with metal-based chemotherapeutic drugs such as palladium has been investigated [258]. The palladium complex has been shown to increase apoptosis in PCa cells. At the same time, chloroquine pre-treatment increased apoptosis via a mitochondria-mediated pathway while decreasing PI3K/AKT/mTOR-related protein expression.

Other inhibitors, such as metformin or fenofibrate, have also been studied. Metformin is an oral biguanide frequently used to treat type 2 diabetes, well known by mTOR inhibition. While it is expected that this should increase autophagy, metformin actually inhibits autophagy by blocking Beclin-1 [259]. A recent study found that metformin may increase autophagy in PCa cells and inhibit cell proliferation by activating the AMPK pathway, which is associated with increased autophagic flux and apoptotic activity [260]. These findings are consistent with autophagy’s multifaceted effects on cell viability. Fenofibrate has also been employed as a modulator of autophagy in docetaxel-resistant PCa cells. It has been established that combining fenofibrate with docetaxel increases taxanes sensitivity and induces PCa autophagy [261]. The presence of fibrates disrupts the cell’s energy metabolism, resulting in a state of metabolic stress. These results point to autophagic induction as an attempt to rebalance energy levels or hasten cell death [261].

Unfortunately, the clinical use of these drugs could be challenging. For example, a high dose of pantoprazole, a drug that has been considered as a possible chemotherapy adjuvant, was shown to be beneficial in blocking autophagy and preventing taxanes resistance in a variety of solid tumors [262]. However, despite the medication being well tolerated by the cohort, the resultant clinical activity was insufficient to justify further testing, as demonstrated by the phase II PANDORA trial [263].

## Figures and Tables

**Figure 1 ijms-24-02939-f001:**
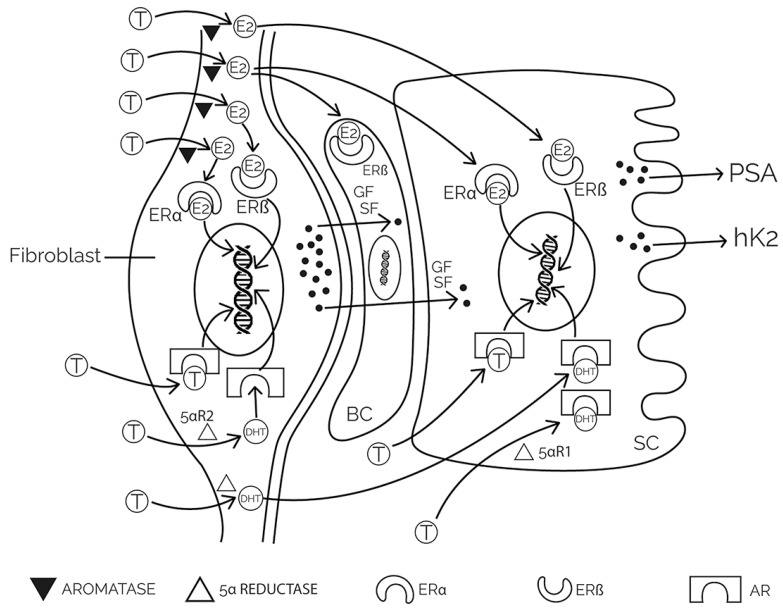
Stroma–epithelium interaction. E2 = estradiol; T = testosterone; 5α-DHT = 5α- dihydrotestosterone; GF = growth factor; SF = survival factor; PSA = prostate specific antigen; hK2 = human glandular kallikrein; BC = basal cell, SC = secretory cell. (Drawing: Pejčić T.).

**Figure 2 ijms-24-02939-f002:**
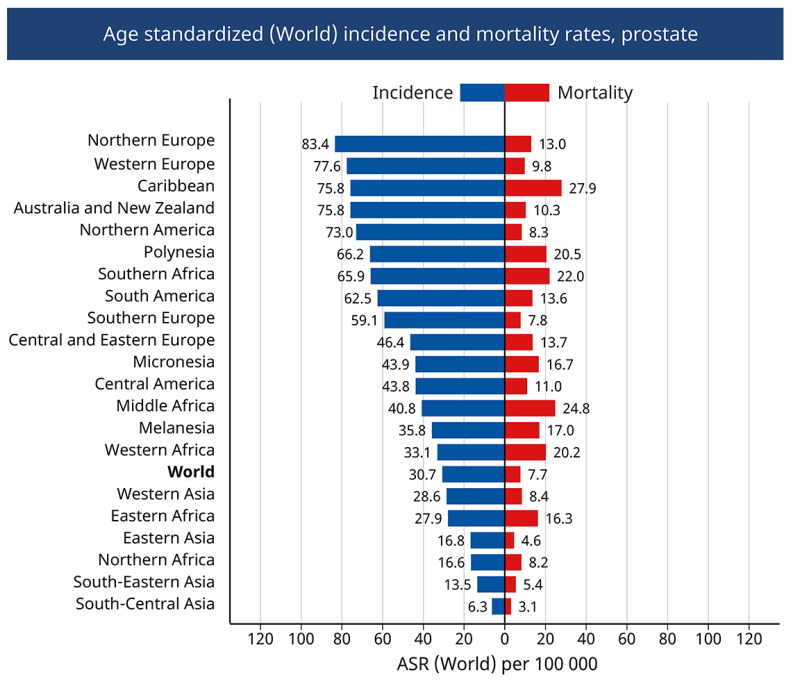
Incidence and mortality rates by age group for PCa in 2020. Source: GLOBOCAN 2020.

**Table 1 ijms-24-02939-t001:** Selected nonsteroidal inhibitors of androgen receptors.

The Name of the Antagonist	Affinity for Androgens Receptors (IC50) *
Hydroxyflutamide	700 nM
Bicalutamide	160 nM
Apalutamide	93 nM
Enzalutamide	86 nM
Darolutamide **	11 nM

* The lower the IC50 (mean inhibitory concentration), the higher the affinity; ** Darolutamide does not cross the blood-brain barrier.

**Table 2 ijms-24-02939-t002:** Inhibition of transduction mechanisms related to androgen receptors under the influence of small molecules.

A Drugs Class	A Molecular Target	Drug Example	Mechanism of Resistance
Androgen synthesis, serum testosterone levels	Gonadotropin synthesis in the adenohypophysis. GnRH agonists	Goserelin	Intratumoral androgen synthesis, AR gene amplification
Androgen synthesis	Intratumorally CYP17	Abiraterone	Use of glucocorticoid receptors and glucocorticoids
Activation of testosterone to dihydrotestosterone	Steroid 5α-reductase	Dutasteride, finasteride	Changing the 5α-reductase isoform or using testosterone or adrenal androgens
AR inhibitors	Dihydrotestosterone binding to monomeric AR	Steroids: estrogens, cyproterone	AR inhibitors
AR dimerization	Androgen receptor	Stilbene, resveratrol	AR gene mutation or expression of ligand-independent splice variants
AR phosphorylation	Phosphorylation site AR kinase:		AR phosphorylation
Translocation of AR in the nucleus	AR	Enzalutamide	Expression of AR splice variants that translocate to the nucleus in the absence of androgens
Binding of nuclear AR binding sites to DNA or to coactivators	AR and coactivators	DNA binding site: 4-(4-phenylthiazol-2-yl)	Changes in coactivator and corepressor balance and relative receptor affinity due to AR gene mutation
Effects of proteins that are influenced by AR (signaling pathways)	Androgen-responsive signaling molecules	morpholine	Activation of alternative salvage pathways that stimulate AR signaling pathways
Stability and degradation of AR	HSP90, serine proteases, calpain and caspases	Different	HS chaperone system becomes redundant, inhibition of AR proteolysis

## Data Availability

Not applicable.

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
