# Peer review of "Mechanisms of Prostate Cancer Cells Survival and Their Therapeutic Targeting"

_ijms, 2023, doi:10.3390/ijms24032939_

Round 1

Reviewer 1 Report

I have read with interest in the manuscript, “Mechanisms of prostate cancer cells survival and advances in their therapeutic targeting” which has been submitted to “Molecular Oncology”. The topic of this paper is important due to how many men are affected by this disease around the world. First of all, I congratulate this work, it is well-written manuscript with the correct explanation of the mechanisms of prostate cancer formation and a detailed description of currently available treatment, as well good conclusions. In addition, tables and figures have been included to facilitate reading the text.

Minor suggestion:

In the mechanisms of formation, the role of diet as a factor influencing the occurrence of PCa has been described. It would be worth considering whether to include in the text a fragment about the impact of diet on the intestinal microbiota, intestinal diseases and further on the occurrence of PCa. This is a new line of research into the influence of various factors on prostate disease and is just worth mentioning in the manuscript.

Author Response

Answer: We thank you for the review and the suggestion to include the impact of microbiota on PCa. In this sense, we have added chapter 3.1.3. Microbiome influence.

All changes are highlighted in green, including in the list of references.

Reviewer 2 Report

The authors have written an overview on the prostate cancer pathogenesis, mechanisms of therapeutic resistance and touched on possible treatments to overcome resistance. They have included an enormous amount of information and it is mostly well-written.

However, too much text is spent describing the prostate and basics of epidemiology of prostate cancer while not enough is spent on discussing the main aim of the article: mechanisms of resistance and advances in their targeting.

 - The introductory sections need to be condensed, while more time needs to be spent on mechanisms of resistance of therapies to address them with an expansion beyond AR targeting agents. For example, there is no mention of AKT inhibition or CD4/6 inhibition, despite the emerging data/ongoing phase III data for these strategies. 

Some more specific comments below:

Instead of ethology, I believe the authors intend to say aetiology/etiology

Use of "etc" too vague/casual, please expand or simply end the statement if adequate information is provided (e.g. lines 42, 58 and elsewhere).

Need references for several statements including the rates of PCa mortality (line 40), statement on the "epidemic" of prostate cancer and relation to animal husbandry (line 43), differences in the incidence of PCa in the East vs. the West (line 45), statements in lines 51-52, 54-60.

Line 67 - PCa already defined earlier in the article.

Line 128 - remove "rarely" to make the sentence more clear

Line 141 - overstating the strength of data. Suggest attenuating statement to say nutritional factors "appear" to play a role.... 

I'd like to see section 3 condense as it is very lengthy and basic when the aim of the article is to focus on mechanisms of PCa cell survival and therapeutic targeting.

Section 4 - again needs to be consolidated/shortened. No need to speak about the history of PCa and to repeat comments on incidence. 

Line 230 - not correct as written, enza/apa/abi are effective but acquired resistance occurs necessitating the need for additional treatment. The mechanism of drugs is also over-simplified (e.g. abi affects androgen synthesis in other sites, not just tumoural tissue). 

Section 4.2 - need to mention LHRH antagonists (e.g. degarelix)

Line 271 - "exacerbation phase" is not typical terminology. Abiraterone is even more effective prior to castrate-resistance so this is a misleading sentence (Stampede, latitude data). 

Line 277 - mechanism of what? Therapeutic targeting of the AR pathway?  Perhaps these sections (4.2 and 4.3) should be subsections of 4.1 if they're a continuation of that section.

Section 4.4 - ADT may be used with radiotherapy in localised disease as well, not only after recurrence. 

Throughout, need to delineate between antiandrogens and ADT/LHRH agents or use a different term to refer to both (e.g. androgen pathway inhibitor agents). 

331 - High-risk forms of what? Cells or cancers? Nearly all cancers will progress to CRPC if given ADT for an adequate period of time. 

Table 2 - please clarify what the "C group", "S group" and M+S group mean.

Line 392 section - need references. Tumour size doesn't necessarily increase likelihood of developing CRPC. There are many other factors (e.g. presence of aggressive variant prostate cancer alterations). 

396 - the wording of this sentence is unclear. Seems to say PSA is a good biomarker but I suspect the opposite is what the authors meant.

Section 6 - the shift in using multiple agents upfront for metastatic prostate cancer needs to be better emphasised. ADT monotherapy is not the standard of care in most metastatic patients. 

Author Response

Answer: We thank you for the review and all suggestions. In this sense, we have rearranged some chapters of the text in accordance with your suggestion, we have added two new chapters that we believe should have been added (4.2.4. Changes in cofactors for AR and 4.2.5. Creation of different variants of AR), we have corrected Table 2, in which the data from one of our previous papers remained unchanged, and we have tried to make additions throughout the text according to your suggestions. All changes are highlighted in green, including the list of references.

Reviewer 3 Report

The review article is basically well-written and easily understandable. 

To improve the manuscript, the reviewer recommends revising the article as below:

1) estrogen and Oestrogens should be unified.

2) Description in line 210 "AR mutations, amplification and overexpression... prostate carcinogenesis. " should be more deeply described.

3) The reviewer thinks that description in line 230-232 "However, all these drugs have not been shown to be sufficiently effective in CRPC..."is correct.  At least enzalutamide and abiraterone are effective against CRPC according to randomized trials. 

4) Molecular weight of AR is not 11kDa but 110kDa. Please check the description in line 240.

5) Please present some description about LHRH antagonist after the description about LHRH agonists in line 257-260.

6) The reviewer thinks that there need some references in line 275.

7) BRCA1, BRCA2, ATM in line 472 should be described in italic characters.

Author Response

Answer: We thank you for the review and all suggestions.

In this sense, we have unified estrogens into estrogens, we corrected AR molecular weight, we presented BRCA1, BRCA2 and ATM in italic, and we added a number of new references.

Specifically, "AR mutations, amplification and overexpression... prostate carcinogenesis" are in details given in subsequent subchapters 4.2.1 Point mutations of the AR gene; 4.2.2 Amplification of AR and CYP17; 4.2.3 Changes in androgen biosynthesis; 4.2.4 Changes in cofactors for AR; and 4.2.4 Creation of different variants of AR.

After the statement "However, all these drugs have not been shown to be sufficiently effective in CRPC...", we have added sentence "In other words, they are effective but acquired resistance occurs necessitating the need for additional treatment."

Finally, there is a new sentence about LHRH antagonist "Other drugs with the same target are GnRH antagonists, such as degarelix. In contrast with the former drugs, the latter immediately decrease the secretion of LH without its temporary surge"

All changes are highlighted in green, including in the list of references.

Round 2

Reviewer 2 Report

A great deal of work has gone into this edit and there is a lot of detail in this review. Ideally, as per usual practice, each point by the reviewer would be addressed in the author's response as the paper is now very long and it's difficult to find the lines that were pointed out for improvement since the line number are very different now.

There are still some major issues with the content however, so I have not gone point by point through them. Major issues below:

1. Rather than making revisions to the imprecise/inaccurate sections, some of the reviewer's comments have simply been added into ( ) adjacent to the areas of issue without consideration to the flow of the writing/syntax.

 - e.g. line 498 (ADT may 498 be used with radiotherapy in localised disease as well, not only after recurrence) has just been stuck into this paragraph and doesn't really make sense.

 - Similarly, the point about LHRH (GnRH) antagonists should be incorporated into the text not just dropped as an extra sentence with inaccurate/redundant wording.

 - And again lines 271-272, an additional statement has just been added without amending the problematic statement (However, all these drugs have not been shown to be sufficiently effective in CRPCa, when the use of cytotoxic agents such as taxanes becomes necessary).

2. Other inaccurate statements still remain at various points

- e.g. line 372 "The CRPCA may develop even when androgen levels are at low (castration) levels". This is the very definition of CRPC, that it develops in castrate conditions.

3. The biggest issue, however, is that this review is entitled "Mechanisms of prostate cancer cells survival and advances in their therapeutic targeting" but rather than covering these topics comprehensively, it spends a huge amount of text going over the basic epidemiology of prostate cancer and historical therapies, an improved amount of text on mechanisms of survival since the revision but very little text on advances in their therapeutic targeting. 

 - The treatment section (seemingly the section that is meant to focus on advances in therapeutic targeting) lacks any data from the PARP inhibitors in combinations in CRPC in non HRD mutant patients (Propel, Magnitude, Talapro Phase II and III studies). It also lacks any mention of the Phase III AKT inhibitor studies. These therapeutic advances (rationale for AKT inhibition and/or PARP inhibition in HRD intact tumours)  could also be discussed in the resistance mechanism sections.

 - The rationale for triplet therapy (and even doublet) therapy mechanistically isn't discussed either (e.g. targeting resistant subclones upfront)

 - Either the up to date clinical data need to be presented or the mechanistic rationale. Otherwise this review is just a generic review with no real aim.

Overall - I'd suggest the authors need to refine the aim of this review or reconsider the title to better reflect its content if they don't wish to cover the advances in therapeutic targeting. If they truly want to address recent therapeutic advances, several of the epidemiological/historical references need to be condensed prior to adding the current therapeutic data.

 - The abstract actually is not clear in the specific aim of this review and perhaps that needs to be reviewed. Namely the following sentence doesn't really define a clear goal as it is currently written -- "New knowledge about the biology of the so-called castration-resistant PC and the way it adapts to therapy will enable the development of new drugs, whose goal is to prolong the survival of patients with this stage of the disease, which will be discussed in this review."

 - The brief discussion of PARP inhibitors only in its HRD context in the conclusion doesn't make sense and doesn't provide a robust conclusion to the text

Author Response

Thanks for the helpful comments. All changes, including new references, are highlighted in green.

Your note:
"1. Rather than making revisions to the imprecise/inaccurate sections, some of the reviewer's comments have simply been added into ( ) adjacent to the areas of issue without consideration to the flow of the writing/syntax.
- e.g. line 498 (ADT may 498 be used with radiotherapy in localized disease as well, not only after recurrence) has just been stuck into this paragraph and doesn't really make sense."
Our correction:
The sentence have been deleted (lines 506-507).

Your note:
"- Similarly, the point about LHRH (GnRH) antagonists should be incorporated into the text, not just dropped as an extra sentence with inaccurate/redundant wording."
Our correction:
The new paragraph on GnRH antagonists has been added (lines 305-314).

Your note:
"- And again lines 271-272, an additional statement has just been added without amending the problematic statement (However, all these drugs have not been shown to be sufficiently effective in CRPCa, when the use of cytotoxic agents such as taxanes becomes necessary) .
Our correction:
A broader explanation has been added, and the entire paragraph has been rearranged (lines 255-266).

Your note:
"2. Other inaccurate statements still remain at various points
- e.g. line 372 "The CRPCA may develop even when androgen levels are at low (castration) levels". This is the very definition of CRPC, that it develops in castrate conditions."
Our correction:
The sentence has been deleted.

Your note:
"- The brief discussion of PARP inhibitors only in its HRD context in the conclusion doesn't make sense and doesn't provide a robust conclusion to the text."
Our correction:
The entire paragraph on PARP inhibitors has been amended and moved to section 4 (lines 267-279).

Your note:
"The biggest issue, however, is that this review is entitled "Mechanisms of prostate cancer cells survival and advances in their therapeutic targeting" but rather than covering these topics comprehensively, it spends a huge amount of text going over the basic epidemiology of prostate cancer and historical therapies, an improved amount of text on mechanisms of survival since the revision but very little text on advances in their therapeutic targeting."
Our correction:
We removed words “advances in” from the title, so now it is  “Mechanisms of prostate cancer cells survival and their therapeutic targeting”. We hope that now it sounds more realistic regarding the manuscript text.